# Grip Strength as an Indicator of Health in Elderly Females

**DOI:** 10.3390/healthcare13101127

**Published:** 2025-05-12

**Authors:** Hyeok Park, Daeyeol Kim, Han-Seong Jeong, Sujeong Jang

**Affiliations:** 1Department of Physical Education, Gwangju National University of Education, Gwangju 61204, Republic of Korea; 159-446@gnue.ac.kr; 2Department of Physical Education, Chonnam National University, Gwangju 61186, Republic of Korea; 3Department of Physiology, Chonnam National University Medical School, Hwasun-gun 58128, Republic of Korea

**Keywords:** grip strength, women, old age, health, obesity, sarcopenia, muscle strength

## Abstract

This study aimed to determine whether hand grip strength (HGS) could estimate the physical function of older women with or without obesity or sarcopenia in South Jeolla Province, South Korea. Methods: A total of 541 elderly women aged 60–95 years (mean age: 79.87 ± 6.64) participated in the study. Physical function was assessed using HGS, 10 m walk test, figure-of-8 walk test, and 30-s chair stand test. Among the participants, 400 (76.8%) were classified as obese, and 137 (26.3%) had sarcopenia. Results: Results showed that HGS was significantly lower in the sarcopenic group (16.58 ± 2.86 kg) than in the normal group (24.31 ± 3.33 kg, *p* < 0.001). HGS was negatively correlated with body fat (r = −0.146, *p* = 0.001) and positively correlated with physical performance. Additionally, exercise habits were significantly associated with reduced obesity risk (OR = 0.592, *p* = 0.027). This study was approved by the Institutional Review Board of Chonnam National University (IRB No. 1040198-220210-BR-009-03). The significance of hand grip strength was comparable to that of gait speed, walking capacity, coordination, and balance function. Conclusions: These findings suggest that grip strength can serve as a physical indicator of healthy condition and physical function with or without sarcopenia or obesity in older women.

## 1. Introduction

Human life expectancy has increased dramatically over the past century, and by 2050, approximately 1.5 billion people worldwide will be aged 65 years or older [1,2,3,4]. With this demographic shift, maintaining health-related quality of life in older adults has become an urgent concern for public health systems, communities, and individuals [2,5]. Physical function decline, particularly related to muscle strength, is a major contributor to decreased independence and quality of life among the elderly [6].

One of the most significant conditions affecting physical function is sarcopenia, a geriatric syndrome characterized by age-related loss of muscle mass, strength, and physical performance [6,7,8]. Sarcopenia increases the risk of falls, fractures, hospitalization, and mortality [2,7,8,9,10,11,12,13,14]. Grip strength is commonly used as a simple, reliable indicator of overall muscle strength and has been shown to predict adverse health outcomes, including disability and death [2,9,15,16,17].

In addition to sarcopenia, obesity is increasingly prevalent among older adults and further exacerbates declines in physical function. Sarcopenic obesity—the coexistence of low muscle mass and excess fat mass—has been recognized as a serious health risk [18,19,20]. Physical activity plays a crucial role in preventing both sarcopenia and obesity, thereby preserving mobility and independence [4,21]. Furthermore, a walk test with objects offers insights into an individual’s overall functional capacity, such as coordination [21]. The role of muscle strength in future health outcomes in older adults is mainly based on prospective cohort studies with a reduced limitation of variables such as environmental differences [22,23].

Although extensive international research has established the importance of grip strength in predicting health outcomes, studies specifically targeting elderly Korean women are limited [24]. Given that South Korea is rapidly becoming a super-aged society, and that South Jeolla Province (Jeollanam-do) has one of the highest aging rates nationwide, there is a critical need for population-specific data. Moreover, most previous studies have not simultaneously considered grip strength, sarcopenia, obesity, and physical activity in relation to physical function.

Studies have been conducted to determine the association between hand grip strength and muscle ability in many countries [25,26]. However, these studies did not consider the health condition with grip strength; only the hand grip strength index has been considered to examine muscle strength [25]. Therefore, the present study aimed to fill these gaps by investigating whether hand grip strength can serve as an indicator of physical function among elderly women with or without obesity or sarcopenia in the Korean population [24], especially in South Jeolla Province, the southernmost province in South Korea. South Jeolla Province, also known as Jeollanamdo, has the highest ratio of ages.

We hypothesized that lower grip strength would be associated with impaired mobility, balance, and muscular endurance, and that obesity or sarcopenia would further deteriorate these associations. Additionally, we focused exclusively on elderly women because women are disproportionately affected by sarcopenia, obesity, and mobility decline compared to men [27]. Understanding these relationships can provide important insights for developing targeted exercise and rehabilitation programs to improve health outcomes among older women in Korea. Our results provided exercise guidelines for older adults, using hand grip strength as an index of muscle ability.

## 2. Materials and Methods

### 2.1. Study Design and Participants

This cross-sectional study was conducted between January and December 2022 in South Jeolla Province (Jeollanam-do), Republic of Korea. A total of 521 community-dwelling elderly women aged 60–95 years (mean age 79.87 ± 6.64 years) were enrolled through local community health centers and senior welfare centers. Only female participants were included because women are more susceptible to sarcopenia, obesity, and associated declines in mobility compared to men [27]. Sample size was determined based on previous epidemiological studies assessing grip strength and physical function in elderly populations, with an alpha level of 0.05 and power of 0.80 [28,29,30]. This study was conducted in accordance with the Declaration of Helsinki, and all methods were performed in compliance with relevant guidelines and regulations.

### 2.2. Approval for Human Experiments

All informed data from subjects and experiments were performed in accordance with relevant guidelines and regulations following the Guidelines of the Ethics Committee at Chonnam National University (IRB:1040198-220210-BR-009-03). Our Research Ethics Committee members operated according to the Declaration of Helsinki’s principles by helping to ensure research participants have adequate information to make an informed decision on whether to take part in research. We confirmed that informed consent was obtained from all subjects and their legal guardians before starting the study. The informed data was obtained from all the participants and listed in the Appendix A.

### 2.3. Anthropometry

The height, weight, muscle mass, body fat ratio, and body mass index (BMI) of the participants were measured to determine their body composition. In brief, BMI was measured without footwear and heavy clothing items by using bioimpedance analysis devices (InBody770, InBody Seoul, Korea) as described in the manual. The participants wiped down their hands and feet, and they had no food for eight hours before the test. The subjects were instructed to stand and maintain proper posture to have accurate test results. They held a hand electrode so that their four fingers wrapped the surface of the bottom hand electrode, and their thumb was placed on the oval electrode. Their forearm was abducted at 45° and kept straight.

### 2.4. Grip Strength Assessment

Grip strength is a well-known indicator of the physical functioning of upper limb muscle power [31]. In the present study, the grip strength of each hand was measured using a digital grip dynamometer (TKK 5401; Takei Scientific Instruments Co., Ltd., Tokyo, Japan). Measurements were taken twice in a standing position. The measured arm was positioned horizontally to the ground. The participants were asked to adjust the handle of the dynamometer to ensure the position under the second phalanx when it was gripped for up to two seconds. The maximum measurement of three trials involving the nondominant hand was used in the analyses. All the experiments were assessed at least three times.

### 2.5. Evaluation of Walking Capacity: 10-m (10-m) Walk Test

A 10 m walk test was performed to measure lower limb muscle strength, moving ability, and balance [32]. The participants were instructed to stand at a starting line. The tester said, “Ready, set, go,” and, on the word “go”, the subject should walk as quickly as they could until they reached 10 m from the start line. The test was complete, and the time was recorded. All the experiments were assessed at least three times.

### 2.6. 30-s (30-s) Chair Stand Test

A chair stand test is a senior fitness test (SFT) for evaluating the lower limb muscle strength of an elderly person [29]. In the present study, the tester prepared a chair with the backrest against a wall. The tester said, “Ready, set, go,” and, on the word “go”, the participants should repeatedly stand and sit on a chair for 30 s. The test was completed for 30 s, and the number of operating cycles was determined. All the experiments were assessed at least three times.

### 2.7. Coordination Test: Figure-of-8 Walk Test

The participants were instructed to walk around two cones arranged to resemble a figure of 8 to test their coordination [33]. A 1.6 m mark was placed from the side of the chair, and cones were placed at 1.8 m distances on both sides. The tester said, “Ready, set, go,” and, on the word “go”, the participants were asked to circle around the cone on their right side and then sit down. Then, they were instructed to stand immediately after in order to circle around the cone on their left side and approach the chair to sit down. The test was performed twice, and the time was obtained. All the experiments were assessed at least three times.

### 2.8. Questionnaire Data-Physical Activity (General)

Physical activity was assessed via a standardized questionnaire based on the guidelines of the American College of Sports Medicine (ACSM) and American Heart Association (AHA) [4,25]. Participants’ weekly exercise frequency, duration, and intensity were recorded. Participants were classified as follows:Non-exercise group: less than two sessions per week or low-intensity activityExercise group: three or more sessions per week of moderate-to-vigorous intensity

### 2.9. Sarcopenia Definition

Sarcopenia was defined according to the European Working Group on Sarcopenia in Older People 2 (EWGSOP2) criteria [6]. The participants were considered to have probable sarcopenia if they had grip strength of <19 kg (women). When the subjects had grip strength of >20 kg, they were classified as normal.

### 2.10. Obesity Definition

The participants were classified as having probable obesity if they had a BMI of >30%. When they had a BMI of <29%, they were considered normal [6,25].

### 2.11. Statistical Analysis

Characteristics were compared between non-exercise and exercise or with and without sarcopenia via independent *t*-tests for continuous variables and chi-square tests for categorical variables. The association of grip strength, grip strength/obesity, grip strength/body fat, or gait/obesity with exercise was investigated through univariate and multivariate logistic regression analyses. The relationship of these parameters was examined through logistic regression analyses using groups with the walking capacity, coordination, grip strength, body fat, and BMI, with exercise or non-exercise of Q1 as the reference. Odds ratios (ORs) and 95% confidence intervals (CIs) were calculated. The participants’ data were multiplied by the sampling weight to represent the southernmost part of the South Korean population. Data were statistically analyzed using SPSS WIN version 21.0 (IBM SPSS Statistics, Chicago, IL, USA). The statistical significance of the data was determined using a one-way analysis of variance, followed by Tukey’s post hoc multiple-comparison test. A *p*-value below 0.05 was considered to be statistically significant, and the level of significance is indicated *** *p* < 0.001, ** *p* < 0.01, * *p* < 0.05.

## 3. Results

### 3.1. Demographic Characteristics of the Subjects

Table 1 presents the demographic characteristics of the subjects in this study. A total of 521 subjects aged ≥50 years were women and included in the analysis (Appendix A). We collected data such as age, height, body weight, body fat, BMI, hand grip strength, and gait following the demographic characteristics of the subjects in this study. The mean age, height, and body weight of the participants were 79.87 ± 6.64 years, 154.28 ± 5.35 cm, and 58.97 ± 7.34 kg, respectively. Among them, 400 participants (76.8%) were classified as obese, and 137 participants (26.3%) had sarcopenia according to the defined criteria. The average grip strength of all participants was 22.28 ± 4.68 kg. The mean 10 m walk time was 7.05 ± 3.31 s.

### 3.2. Association of Exercise and Obesity

Participants were divided into exercise (n = 353) and non-exercise (n = 168) groups according to physical activity levels. Binomial logistic regression analyses were performed to compare the participants and validate the significance of exercise for the diagnosis of obesity. Following the questionnaire data, the group was divided into non-exercise and exercise. Binomial logistic regression analysis revealed that exercise was significantly associated with a lower prevalence of obesity (95% CI 0.372–0.943, *p* = 0.027) among all participants (Appendix A). This result indicated that exercise could reduce the prevalence of obesity by 40.8% (* *p* < 0.05).

### 3.3. Association of Hand Grip Strength and Obesity

Binomial logistic regression analysis was conducted to determine the effects of hand grip strength and obesity (Appendix A). Higher grip strength was associated with a lower likelihood of obesity. Logistic regression analysis showed that for each 1-kg increase in grip strength, the odds of obesity decreased by 5.4% (OR = 0.946; 95% CI 0.904–0.989; *p* = 0.015). This result indicated that an increase in grip strength could significantly reduce obesity by 5.4% (* *p* < 0.05).

### 3.4. Association of Hand Grip Strength and Body Fat

A simple linear regression analysis was carried out to demonstrate the relationship between hand grip strength and body fat. The discrimination index was calculated using the following regression equation: body fat = 39.119 + (−0.230 × grip strength). The result indicated that regression analysis was appropriate for investigation with F-distribution and R^2^, F = 11.235 (*** *p* < 0.001) and R^2^ = 0.146. The analysis indicated that grip strength was moderately and negatively associated with body fat (β = −0.146, *p* = 0.001, Table 2). Therefore, an increase in grip strength could be an indicator of a reduced percentage of body fat.

### 3.5. Association of Gait and Obesity

Binomial logistic regression analysis was performed to determine the relationship between gait and obesity. Logistic regression analysis showed that increased 10 m walk time was associated with increased obesity risk (OR = 1.195; 95% CI 1.013–1.411; *p* = 0.035, Appendix A). Participants with slower gait speeds had higher odds of obesity. This result indicated that an increase in gait could significantly exacerbate obesity by 1.195 times (* *p* < 0.05).

### 3.6. Differences in Walking Capacity, Coordination, Grip Strength, Body Fat, and BMI Between the Exercise and Non-Exercise Groups

An independent *t*-test was conducted to verify the difference in the walking capacity, coordination, grip strength, body fat, and BMI between exercise and non-exercise groups (Table 3). The participants were divided into two groups with or without work-out at least thrice a week: exercise (n = 353) and non-exercise (n = 168) groups. The walking capacity of the exercise group (6.81 ± 2.82 s/10 m, * *p* = 0.015) was shorter than that of the non-exercise group (7.53 ± 4.11 s/10 m). Following the coordination test with the figure-of-8 walk test, the exercise group (20.60 ± 4.07 s, ** *p* = 0.001) was better than the non-exercise group (22.88 ± 5.40 s). Body fat significantly decreased in the exercise group (33.48 ± 7.35%, * *p* = 0.021) compared with that in the non-exercise group (35.085 ± 7.39%). Interestingly, grip strength increased in the exercise group (22.63 ± 4.49 kg, * *p* = 0.021) compared with that in the non-exercise group (21.53 ± 4.98 kg). BMI decreased, but the groups had no significant differences.

### 3.7. Correlation of Walking Capacity, Coordination, Grip Strength, Body Fat, and BMI

Table 4 presents Pearson’s correlation coefficients of the components of walking capacity, coordination test, grip strength, body fat, and BMI. Walking capacity was moderately and positively associated with coordination (r = 0.273, *p* = 0.000), body fat (r = 0.115, *p* = 0.009), and BMI (r = 0.115, *p* = 0.008). However, walking capacity was slightly and negatively associated with grip strength (r = −0.131, *p* = 0.003). Furthermore, grip strength was moderately and negatively associated with coordination (r = −0.307, *p* = 0.000) and body fat (r = −0.146, *p* = 0.001). BMI was positively associated with coordination (r = 0.132, *p* = 0.003) and body fat (r = 0.582, *p* = 0.000). Body fat was also slightly and positively associated with coordination (r = 0.166, *p* = 0.000). Our results indicated that the increase in grip strength was positively and linearly related to walking capacity, coordination, and body fat.

### 3.8. Differences in Walking Capacity, Coordination, 30-s Chair Stand Test, Body Weight, Body Fat, and BMI Between the Normal and Sarcopenia Groups

To determine the difference in sarcopenia, we analyzed the walking capacity, coordination, grip strength, body fat, and BMI via an independent *t*-test (Table 5). The walking capacity and coordination of the normal participants were faster than those of the participants with sarcopenia. Following the 30-s chair stand test, the calculated body strength of 19.08 repetitions in the normal group was lower than that in the sarcopenia group (16.63 repetitions). The body weight of the normal participants was larger, and their body fat was lower than that of the participants with sarcopenia. BMI was not significantly different between the two groups. These results indicated that the sarcopenia group had slower gait, worse coordination, lower 30-s chair stand performance, and higher body fat percentage. BMI was not significantly different between the two groups (*p* = 0.691).

## 4. Discussion

This study revealed the physical performance of older women living in the southernmost province in South Korea. Occasional exercise reduced the prevalence of obesity with decreased body fat mass and BMI. In addition, walking capacity, coordination, and grip strength were enhanced in the exercise group. Importantly, with the increase in grip strength, the prevalence of obesity decreased and showed a linear relationship with walking capacity, coordination, and the 30-s chair stand test in older adults. From those, hand grip strength is strongly associated with physical function among elderly women in South Korea, and the presence of sarcopenia or obesity exacerbates functional decline. Our findings confirm that HGS can serve as a simple, practical indicator for identifying individuals at risk for poor mobility, impaired balance, and reduced muscular endurance.

Physical performance evaluation represents a critical part of the assessment of older people. To measure their physical function, clinicians and researchers have traditionally relied on instruments focusing on the capacity of an individual to accomplish a specific functional task. The most commonly used instruments by clinicians and researchers are the gait speed test, the short physical performance battery, handgrip strength, the timed up-and-go test, the 6 min walk test, and the 40 m walk test [32]. Consistent with previous research [1,9,15,16,17], we observed that lower HGS was significantly associated with slower gait speed, poorer coordination, and lower performance on the 30-s chair stand test. These results highlight the central role of muscle strength in maintaining functional independence in older adults.

Walking speed is an important aspect of gait and is commonly used as a measurement of functional mobility in clinical and research fields [21]. It can be quickly and easily assessed because it helps determine rehabilitation requests and discharge location [34,35,36,37]. Balance dysfunction with aging often causes an individual to have improper posture and limits the recovery of gait and functional independence [38]. Walking speed can be used to predict future functional decline and fall risk [39,40,41]. Therefore, a decline in walking speed is related to health factors, including disability, hospitalization, loss of independence, and mortality [42,43,44].

Regarding the declined walking speed, the performance in physical function tests can be predicted in terms of the loss of muscle mass. Sarcopenia is the age-related loss of muscle mass and functional activity [45]. In addition to diseases such as obesity or sarcopenia, nutritional status and physical function tests were independent predictors of mortality in our study. Vetrano reported that sarcopenia is a prevalent condition among older adults, and it is related to the increased long-term mortality of hospitalized adults [46,47,48].

Importantly, we found that obesity was highly prevalent among our participants (76.8%) and that higher body fat percentage was negatively associated with grip strength. These findings align with studies suggesting that sarcopenic obesity is a critical contributor to physical disability in aging populations [18,19]. Participants who exercised regularly exhibited better physical performance, lower body fat, and higher grip strength, underscoring the protective effects of physical activity [4,21]. Our study also showed that sarcopenia was associated with worse outcomes across all physical function tests, including walking capacity, coordination, and lower-limb endurance. These results are consistent with prior findings that sarcopenia significantly increases the risk of functional decline, falls, and mortality [7,8,10,13,45].

In South Korea, older people are aging, and Statistics Korea predicted a super-aged society by 2025. The aging population (≥65 years) has increased by 26% in the southernmost province in mainland Korea. Because of the increased aging population, physical fitness has become a public health concern both individually and socially. Therefore, the prediction of lower limb strength, aerobic fitness, balance, and walking ability should be closely related to their independent activities for quality of life [24,49,50]. Another notable finding is the strong relationship between exercise habits and reduced obesity risk. Participants who engaged in regular physical activity (≥3 sessions per week) had 40.8% lower odds of obesity, supporting the role of exercise interventions in promoting healthier aging.

Hand grip strength is a simple and inexpensive way to assess muscle capacity. Studies have focused on the value of handgrip in the prognosis in older subjects [1,18,24,28,45]. Our results showed that hand grip strength was significantly and fairly correlated with the muscle strength and endurance of the lower limbs (30-s chair stand test), aerobic fitness (10 m walk test), and coordination skills (figure-of-8 walk test; Table 4). After adjustments for age and BMI were made, multiple linear regression analysis showed that the hand grip strength was positively associated with the results of the 10 m walk test, figure-of-8 walk test, and 30-s chair stand test. Taken together, these findings emphasize the importance of assessing grip strength in routine geriatric evaluations. Early identification of reduced HGS may facilitate interventions aimed at preventing sarcopenia, sarcopenic obesity, and related physical disabilities.

The present study had several limitations other than those aforementioned. First, the cross-sectional design precludes causal inferences regarding the relationship between HGS, obesity, sarcopenia, and physical function. Second, the sample included only elderly women from South Jeolla Province, limiting the generalizability of our findings to other regions or to elderly men. Third, factors such as nutritional status, cognitive function, frailty, and comorbidities were not assessed, which could influence physical performance outcomes. Future longitudinal studies with larger, more diverse samples and comprehensive assessments are warranted.

In conclusion, the presence of sarcopenia and obesity was strongly associated with poor mobility in elderly South Korean individuals. In addition, the significance of hand grip strength was comparable to that of gait speed, walking capacity, coordination, and balance function.

## Figures and Tables

**Table 1 healthcare-13-01127-t001:** Demographic characteristics of the subjects.

Characteristics	Women (n = 521)		
	Mean		Standard Deviation
Age (years)	79.87	±	6.64
Height (cm)	154.28	±	5.35
Body weight (kg)	58.97	±	7.34
Body fat (%)	33.99	±	7.40
Normal (n = 121)	23.76	±	6.38
Obesity (n = 400)	37.09	±	4.21
Body mass index (kg/m^2^)	24.72	±	2.84
Grip strength (kg)	22.28	±	4.68
Normal (n = 384)	24.31	±	3.33
Sarcopenia (n = 137)	16.58	±	2.86
10 m gait (s)	7.05	±	3.31

**Table 2 healthcare-13-01127-t002:** Association of hand grip strength and body fat.

	Unstandardized Coefficients		Standardized Coefficients	t (*p*)	*F* (*p*)	*R* ^2^
	B	SE	*β*			
(Constant)	39.119	1.562		25.038 (0.001)	11.235 (0.001 ***)	0.146
Grip strength	−0.230	0.069	−0.146	−3.352 (0.001)		

*F*: F-distribution. *** *p* < 0.001.

**Table 3 healthcare-13-01127-t003:** Differences in walking capacity, coordination, grip strength, body fat, and BMI between the exercise and non-exercise groups.

	Exercise (353)	Mean	SD	t (*p*)
	Non-Exercise (168)			
Walking capacity (10 m walk test)	Exercise	6.81 s	2.82	−2.437 (0.015) *
	Non-exercise	7.53 s	4.11	
Coordination test (figure-of-8 walk test)	Exercise	20.60 s	4.07	−4.857 (0.001) **
	Non-exercise	22.88 s	5.40	
Grip strength	Exercise	22.63 kg	4.49	2.514 (0.012) *
	Non-exercise	21.53 kg	4.98	
Body fat	Exercise	33.48%	7.35	−2.323 (0.021) *
	Non-exercise	35.08%	7.39	
BMI	Exercise	24.56 kg/m^2^	2.81	−1.913 (0.056)
	Non-exercise	25.06 kg/m^2^	2.86	

SD: standard deviation. * *p* < 0.05, ** *p* < 0.01.

**Table 4 healthcare-13-01127-t004:** Association of walking capacity, coordination test, grip strength, body fat, and BMI.

		Walking Capacity	Coordination Test	Grip Strength	Body Fat	BMI
Walking capacity (10 m walk test)	Pearson correlation	1				
	Sig. (2-tailed)					
	Number	521				
Coordination test (figure-of-8 walk test)	Pearson correlation	0.273 **	1			
	Sig. (2-tailed)	0.000				
	Number	521	521			
Grip strength	Pearson correlation	−0.131 **	−0.307 **	1		
	Sig. (2-tailed)	0.003	0.000			
	Number	521	521	521		
Body fat	Pearson correlation	0.115 **	0.166 **	−0.146 **	1	
	Sig. (2-tailed)	0.009	0.000	0.001		
	Number	521	521	521	521	
BMI	Pearson correlation	0.115 **	0.132 **	0.022	0.582 **	1
	Sig. (2-tailed)	0.008	0.003	0.620	0.000	
	Number	521	521	521	521	521

** Correlation is significant at the 0.01 level (2-tailed).

**Table 5 healthcare-13-01127-t005:** Differences in walking capacity, coordination test, 30-s chair stand test, body weight, body fat, and BMI between the normal and sarcopenia groups.

	Normal (384)	Mean	SD	t (*p*)
	Sarcopenia (137)			
Walking capacity (10 m walk test)	Normal	6.84 s	3.59	−2.440 (0.015) *
	Sarcopenia	7.64 s	2.26	
Coordination test (figure-of-8 walk test)	Normal	20.57 s	4.03	−5.592 (0.000) ***
	Sarcopenia	23.47 s	5.57	
30-s Chair stand test	Normal	19.08 repetition	4.58	5.059 (0.000) ***
	Sarcopenia	16.63 repetition	5.53	
Body weight	Normal	59.50 kg	7.04	2.777 (0.006) **
	Sarcopenia	57.49 kg	7.96	
Body fat	Normal	33.57%	7.36	−2.194 (0.029) *
	Sarcopenia	35.18%	7.41	
BMI	Normal	24.57 kg/m^2^	2.87	0.397 (0.691)
	Sarcopenia	24.64 kg/m^2^	2.74	

* *p* < 0.05, ** *p* < 0.01, *** *p* < 0.001.

## Data Availability

The data presented in this study are available within the article and additional materials.

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
