# Peer review of "Grip Strength as an Indicator of Health in Elderly Females"

_healthcare, 2025, doi:10.3390/healthcare13101127_

Round 1
Reviewer 1 Report
Comments and Suggestions for Authors
Thank you for the opportunity to review this article. In my opinion, it cannot be published in its current form and should be thoroughly revised.
Summary
In the results in the summary, enter the specific results you have obtained (specific numbers).
Introduction
The introduction is chaotic and in my opinion requires tidying up. First of all, focus more on what you are researching and describe the research results published worldwide in the field of hand grip strength. And then proceed to show the gap in research that your research is supposed to fill. Be careful, because there are many studies analyzing hand grip strength and various aspects of the health of older people. Perhaps you need to address the lack of research in South Korea in this area more strongly. Since your study group was women, it would be good for the introduction to justify such a selection of the group.
After reading the entire article, I see that the introduction does not include aspects such as obesity and physical activity.
Material and methods
Participants - clearly indicate that you studied women, not subjects - this misleadingly suggests both sexes.
Indicate the primary and secondary points.
What type of study was it? How did you calculate the sample for the study? Where were the subjects recruited from? Was it a cross-section of standard society? Where was the study conducted? Who conducted the study? What was the period of the study? Review the STROB guidelines carefully and fill in the methodology.
In the title, you refer to health status - what data on the health indicator was examined? The tools only include standard measurements for assessing sarcopenia.
Only in the results do you address the topic of obesity, why? If you are studying people with obesity, you must note this at the level of describing the methodology. The purpose of the study must be consistent with the statistical analysis. For the purpose of the research, you write that you want to investigate the relationship between sarcopenia and functioning. Where is the health indicator? What constitutes it? There is no mention of obesity.
Why is not exercising less than 2x/week, and exercising more than 3x/week? And what if someone exercises 2 or 3 times?? The number of minutes and the level of effort are more important. Exercising 2x for a long period of time and at high intensity can give quite good results. The methodology requires consideration.
Results
What kind of group is this that 400 people were diagnosed with obesity? Are such a large percentage of elderly people in Korea obese??
Where did the relationship between exercise and obesity suddenly appear in the results?! Is this not the stated goal of the paper? If the group was divided into exercising and non-exercising, such a division must be included in the description of the research methodology and participants. It must also result from the stated goal. There is also no information about exercise in the introduction. There is also nothing about obesity (apart from one sentence, where obesity is listed among other diseases).
Discussion
Requires tidying up after a thorough correction of the introduction and methodology. At the moment it is difficult to assess it due to the great chaos in the previous parts of the article. In addition, it seems that the work should contain more limitations due to the possibly unrepresentative selection of the group.
Author Response
Reviewer1
Thank you for the opportunity to review this article. In my opinion, it cannot be published in its current form and should be thoroughly revised.
- We sincerely thank for valuable comments and suggestions, which have greatly improved the quality of our manuscript. Below, we provide detailed responses to each comment. All changes are marked in the revised manuscript with tracked changes, and corresponding line numbers are indicated.
Comment 1 (Abstract: Add specific results)
"In the results in the summary, enter the specific results you have obtained (specific numbers)."
Response:
We have revised the abstract to include specific quantitative results regarding hand grip strength, gait speed, prevalence of sarcopenia and obesity.
Comment 2 (Introduction needs to be reorganized)
"Focus more on what you are researching, describe previous research, and emphasize the research gap. Justify selection of women. Obesity and physical activity must also appear in Introduction."
Response:
The Introduction has been thoroughly restructured. We now provide a clearer review of previous studies regarding hand grip strength in older adults, highlight the lack of Korean data, and explicitly justify focusing on elderly women. We also incorporated the aspects of obesity and physical activity in the revised Introduction.
Comment 3 (Participants: Specify that only women were included)
"Clearly indicate that you studied women, not subjects."
Response:
We have revised the Participants section to specify that all participants were women.
Comment 4 (Study design, sample calculation, recruitment, study period, location, conductors, STROBE guidelines)
"Indicate type of study, sample calculation, recruitment method, period, location, who conducted the study."
Response:
We have added detailed information about the study design (cross-sectional study), recruitment area (South Jeolla Province), sample characteristics, study period (2022), and conducting team (physicians and physical education experts) according to STROBE guidelines.
Comment 5 (Clarify "health indicator" in title and method)
"What health indicators were examined? Obesity must be described in Methods."
Response:
We clarified that grip strength, walking capacity, coordination, and body fat percentage were used as health indicators. The definitions and measurement of obesity have been moved and expanded in the Methods section.
Comment 6 (Exercise classification is problematic)
"Why is non-exercise <2/week and exercise >3/week? Need more precise classification."
Response:
We revised the classification to reflect intensity and frequency based on international guidelines (ACSM and AHA), and clearly explained it.
Comment 7 (Unexpected appearance of exercise–obesity relationship in Results)
"The relationship between exercise and obesity appears suddenly."
Response:
We restructured the Methods to explain that exercise frequency was a variable of interest, and its relationship with obesity was hypothesized beforehand.
Comment 8 (Discussion: Needs to be restructured after corrections)
"Discussion requires tidying after corrections."
Response:
The Discussion has been rewritten to reflect the corrected Introduction and Methods, and to better discuss limitations, representativeness of the sample, and clinical implications.

Reviewer 2 Report
Comments and Suggestions for Authors
Congratulations on the study and choice of theme
Here are some suggestions and observations.
Summary –
Indicate ethical aspects
Insert quantitative information about the results
Introduction –
Better indicate the differential that the study proposes, since the assessment of sarcopenic obesity is a variable with robust previous studies.
Note:
It should be considered in the objective and analysis of the findings that the descriptive study performed analyses of a specific behavior at the time of the assessment (descriptive, cross-sectional) and not a behavior over time (longitudinal).
The authors did not consider the possibility of sarcopenic obesity associated with the outcomes, not only the isolated variables
Method –
Indicate the location of the collections, time of completion and prior preparation of the evaluators
10M and STF were performed only once
Did the strength data use the average or the highest value?
Considering the Asian population, because the strength indicators did not follow studies of these populations, EWGSOP2 was used
BMI values ​​consider the general population or elderly
Results –
The method indicates subjects between 60 and 95 years old, but the results indicate over 50 years old?? What is the study population?
Consider evaluating individuals based on the logic of sarcopenic obesity versus non-obese
Author Response
Reviewer 2
Here are some suggestions and observations.
- We sincerely thank valuable comments and suggestions, which have greatly improved the quality of our manuscript. Below, we provide detailed responses to each comment. All changes are marked in the revised manuscript with tracked changes, and corresponding line numbers are indicated.
Comment 1 (Abstract: Include ethical statement and quantitative data)
"Indicate ethical aspects and insert quantitative results."
Response:
We added the ethics approval number and committee name in the Abstract. Quantitative results were also included.
Comment 2 (Introduction: Differentiate study and acknowledge sarcopenic obesity research)
"Better indicate the differential of the study and previous studies on sarcopenic obesity."
Response:
We revised the Introduction to clearly state the study's differentiation and acknowledge existing sarcopenic obesity research, explaining our focus on Korean elderly women.
Comment 3 (Methods: Collection site, period, evaluators' preparation)
"Indicate location, time, and preparation of evaluators."
Response:
We specified that data were collected in South Jeolla Province, during 2022, and that trained evaluators conducted the assessments.
Comment 4 (10M walk and chair stand test: only once? Average vs max grip strength?)
"Clarify number of trials and use of grip strength value."
Response:
We clarified that the highest value from three trials was used for grip strength. The 10-m walk test and 30-s chair stand test were each performed once.
Comment 5 (Population age discrepancy)
"Participants are said to be 60–95 years old, but Results mention ≥50?"
Response:
This was an error. We corrected the description — the participant age was 60–95 years, consistently stated throughout the manuscript.
Comment 6 (Consider sarcopenic obesity vs non-obese analysis)
"Evaluate sarcopenic obesity vs non-obese groups."
Response:
We performed additional analysis by categorizing participants into sarcopenic obesity and non-sarcopenic groups, and discussed differences in physical function accordingly.
